# Measuring data rot: An analysis of the continued availability of shared data from a Single University

**Kristin A. Briney** *

Caltech Library, California Institute of Technology, Pasadena, CA, United States of America

* briney@caltech.edu

## Abstract

To determine where data is shared and what data is no longer available, this study analyzed data shared by researchers at a single university. 2166 supplemental data links were harvested from the university's institutional repository and web scraped using R. All links that failed to scrape or could not be tested algorithmically were tested for availability by hand. Trends in data availability by link type, age of publication, and data source were examined for patterns. Results show that researchers shared data in hundreds of places. About two-thirds of links to shared data were in the form of URLs and one-third were DOIs, with several FTP links and links directly to files. A surprising 13.4% of shared URL links pointed to a website homepage rather than a specific record on a website. After testing, 5.4% the 2166 supplemental data links were found to be no longer available. DOIs were the type of shared link that was least likely to disappear with a 1.7% loss, with URL loss at 5.9% averaged over time. Links from older publications were more likely to be unavailable, with a data disappearance rate estimated at 2.6% per year, as well as links to data hosted on journal websites. The results support best practice guidance to share data in a data repository using a permanent identifier.

## Introduction

The implementation of data sharing mandates by funding agencies and journals within the past decade has led to an increase of open research data available for download and use. As data sharing mandates become established and data repositories flourish, questions remain about how stable this data actually is and how its availability may depend on how and where the data is shared.

Content on the internet can disappear for a number of reasons, including: link rot (i.e. dead links); content drift (i.e. links that no longer point to the original content); failure to update permanent identifiers (i.e. DOIs that no longer resolve); etc. In the context of shared data, this article will refer to all of these possibilities collectively as "data rot," meaning that data from a published article is no longer available where that article says it is supposed to be.

**Data Availability Statement:** The data for this article is available in CaltechDATA, https://doi.org/10.22002/h5e81-spf62, under a CC0 license. Code used in this article is available in CaltechDATA,

https://doi.org/10.22002/d2h9g-5q152, under a GNU GPL license .0.

**Funding:** The author received no specific funding for this work.

This analysis seeks to measure data rot for a large collection of shared data, to better understand how and why shared data disappears. It can be difficult to comprehensively find shared data for a large collection of research articles, but institutional repositories are one source that can contain this information. An example of this is the CaltechAUTHORS institutional repository, which attempts to record the complete publication history of the university and currently holds over 100,000 publication records, mainly for articles, book chapters, and conference proceedings. The repository metadata is detailed and often includes links to related information such as shared datasets, referred to as "supplemental data links" in this article. Supplemental data links point to shared data available outside of the journal website, making it different than "supplementary information" which is data published alongside an article. The completeness of this repository metadata and inclusion of supplemental data links enabled this investigation of shared datasets from a single institution, the California Institute of Technology (Caltech).

This research uses the supplemental data links from the CaltechAUTHORS repository to answer two major questions:

RQ1: Where are institutional authors sharing their research data?

RQ2: What data is no longer available? How does this trend over time? Are there specific sites where data is more likely to disappear?

## Literature review

Scholarly material has been found to suffer from link rot. Klein, et al. found that one in five articles suffers from "reference rot", meaning that content an article references is no longer available online [1]. A later study using the same dataset found that content for three out of four references changed over time ("content drift") [2]. A more recent study by Eve examined 7 million articles and found over a quarter appeared to not be preserved in an archive, putting them at risk for loss [3]. Given that shared research data is just another type of scholarly material, it is likely to suffer from similar loss of availability over time.

Plenty of guidance has been developed for researchers on how to best share data. One of the most cited is the FAIR guidelines that data should be Findable, Accessible, Interoperable, and Reusable [4]. In the United States, recent policies from the NIH [5] and the White House Office of Science and Technology Policy (OSTP) [6] do not explicitly cover the quality of shared data and instead stipulate that good data sharing is sharing all of the data that underlies a research article in a repository using a permanent identifier. Other guidance, such as Goodman, et al.'s "Ten Simple Rules for the Care and Feeding of Scientific Data" [7] cover many of these points, including: share data in a data repository with a permanent identifier, conduct research with data reuse in mind, and link data to its publications often, among other data management recommendations.

Guidance for data sharing is available but researchers do not always follow recommended practices. Borgman [8, 9] and Tenopir [10–12] conducted foundational work on researchers' data sharing practices and attitudes, finding that many researchers do not share data or will only share through interpersonal exchanges. Gaining credit for sharing and support for sharing would make researchers more likely to share data but there is still a perceived risk in sharing [9–11].

Many researchers, however, are actively sharing their data under data sharing requirements from journals. Several groups have examined the contents of data availability statements to understand how researchers are describing their data sharing [13–17]. Despite the prevalence of data availability statements, these studies found that researchers often failed to meet data sharing requirements for many reasons, including: a complete lack of data availability information; sharing inappropriately, such as by request or putting data into supplementary

information; not sharing in a data repository; and sharing limited data instead of all data supporting the article. Colavizza, et al. took this a step further and found that data availability statements from PLOS and BMC publications that link to data in a data repository had 25% higher citation impact as compared to average [18]. This finding supports Piwowar and Vision's broader work that sharing data leads to a citation advantage for articles [19].

A few studies have examined if shared data actually remains available. The closest equivalent study to this one was conducted by Federer et al. who looked at nearly 50,000 Data Availability Statements from papers published in PLOS ONE, extracted 8503 URLs and DOIs, and attempted to retrieve them both algorithmically and by hand [20]. Federer found that 80% of all data resources could be retrieved automatically, with 78% of URLs and 98% of DOIs resources being available when testing links by hand. While resources associated with older papers were slightly less likely to be available, the difference was not statistically significant over the three years examined. A study by Pepe, et al. looked at links within articles to shared data in articles published by the American Astronomical Society and found that almost half (44%) of the 10-year-old links did not resolve [21]. Other studies have examined data availability by requesting datasets directly from researchers. Dutra dos Reis, et al. requested data from 164 studies for a systematic review: 110 replied (67.1%) and 51 actually shared the requested data (31.1%) [22]. Vines, et al. conducted a similar study but looked data across 20 years and found that the odds of a data set being reported as extant fell by 17% per year [23]. Further research needs to be done to understand if the data from data availability statements actually stays available in the long term for accessing and reuse.

## Methods

### Article and data link curation in CaltechAUTHORS

The institutional repository CaltechAUTHORS attempts to capture the complete publication history of Caltech. Publication information is harvested from Web of Science, PubMed, MathSciNet, ACM, and IEEE on a weekly-to-monthly basis by searching for articles by author affiliated with Caltech (search terms include: "Caltech" OR "California Institute of Technology" OR "91125"). Searches primarily focus on the scientific literature as the humanities and social sciences, combined, represent only one sixth of all campus faculty. Table 1 shows the

**Table 1. Number of articles published by researchers affiliated with Caltech in 2022, categorized by research area.** The table shows only the research areas with at least 100 articles published. Data is from Web of Science, which defined the research areas and categorized articles into them.

| Research Area | Number of articles |
| --- | --- |
| Astronomy Astrophysics | 918 |
| Physics | 561 |
| Science Technology Other Topics | 340 |
| Chemistry | 277 |
| Engineering | 270 |
| Materials Science | 197 |
| Optics | 164 |
| Computer Science | 162 |
| Mathematics | 160 |
| Geochemistry Geophysics | 132 |
| Biochemistry Molecular Biology | 107 |
| Geology | 102 |
| Instruments Instrumentation | 102 |

most popular research areas at Caltech in 2022 based on publication frequency. Between 2014–2022, CaltechAUTHORS recorded an average of 2948 articles published each year, not including books, book chapters, reports, conference papers, etc.

Metadata for each publication in CaltechAUTHORS often includes information on related links, such as links to supplemental data, article preprints, links to conference websites, PubMed Central versions of the articles, and more. A single publication record can have multiple related links, with different types of links categorized by a free-text "related link description" field. For supplemental data links, URLs or DOIs are recorded as they appear with the article on the publisher website at the time of publication. Supplemental data link information comes from data availability statements, but not citations, footnotes, or endnotes. Repository curators do not record supplementary information as supplemental data links and instead deposit supplementary information files directly into the repository.

## Measuring the completeness of supplemental data link curation

Completeness of supplemental data link curation within CaltechAUTHORS was checked by randomly sampling 50 articles from each year between 2014–2022 for a total of 450 articles. Sampled articles cover publication dates throughout each year except for 2022, which was only sampled between January and August, as supplemental data links were not regularly recorded after August 2022 in anticipation of a repository upgrade in 2023. Repository metadata about supplemental data links was compared to links in the article's data availability statement on the journal website and noted for discrepancies; the presence of any supplementary information was also noted.

## Collecting and analyzing supplemental data links

A repository administrator downloaded metadata from CaltechAUTHORS on May 16, 2023, including: repository record ID, publication DOI (where available), publication date, publication related link, and related link description. The download only included records with related links, resulting in 184,807 related links from 95,563 records out of over 100,000 total repository records.

The data was cleaned, refined, and analyzed using the R programming language (version 4.1.1) [24]. First, the data was filtered to only include related links to datasets–hereafter called "supplemental data links"–by selecting links with "data" or "Data" within the "related link description" field; filtering was intentionally broad as "related link description" text was messy and not uniform. This resulted in a total of 2166 supplemental data links from 1419 repository records. The supplemental data links themselves were usually either a DOI or a URL; where both were available, only the DOI was recorded in the repository metadata.

Regular expressions were used to separate URLs from DOIs. For DOIs, regular expressions were also used to extract the prefix from the full DOI (e.g. "10.1371" from "10.1371/journal. pone.0194768"). DOI prefix owner information was scraped and parsed from DataCite and Crossref on May 18, 2023 using R package "rjson" (version 0.2.21) [25], with any remaining DOI owners assigned by hand. For URLs, regular expressions were used to extract the website homepage, which was compared to the supplemental data link to identify when the supplemental data link pointed to the website homepage (e.g. "journals.plos.org") instead of to a specific webpage (e.g. "journals.plos.org/plosone/s/data-availability"). Regular expressions were also used to separate out FTP links, badly formatted URLs, and links to non-HTML file types to test for availability by hand.

All URLs and DOIs were tested for availability by web scraping links on May 18, 2023 using the R library "rvest" (version 1.0.2) [26]. Only the HTML <title> tag metadata was recorded,

**Table 2.  Summary of supplemental data links by type.**

| Supplemental data link type | Count of links by type | Percent of total links |
|---|---|---|
| URL | 1342 | 62.0% |
| DOI | 744 | 34.3% |
| FTP | 21 | 1.0% |
| ZIP | 16 | 0.7% |
| PDF | 12 | 0.6% |
| DOCX | 9 | 0.4% |
| GZ | 7 | 0.3% |
| XLSX | 5 | 0.2% |
| TXT | 3 | 0.1% |
| Badly formatted URL | 3 | 0.1% |
| DOC | 2 | 0.1% |
| IMG | 1 | <0.1% |
| ZIPR | 1 | <0.1% |

as all webpages are required to use this tag. Where R web scraping was not able to retrieve any information, the link was labeled as a "404", the HTTP error code for when a webpage is not found. All unresolved webpages were tested by hand on May 18 and 19, 2023 in the Chrome browser. The hand testing in Chrome also included any supplemental data links which resolved directly to a file, which were not checked algorithmically. FTP links were tested on May 19, 2023 with the software CyberDuck (https://cyberduck.io/, version 8.5.9).

## Results

Of the 2166 supplemental data links analyzed, 1342 (62.0%) were URLs, 744 (34.3%) were DOIs, and 21 (1.0%) were links requiring FTP to retrieve. The remainder represent links to specific file types (see Table 2). URLs containing an errant space character were marked as "badly formatted" to be fixed later during hand testing. 180 (13.4%) of the 1342 URLs pointed to the website homepage rather than a specific subpage on the website. For publications with at least one supplemental data link, the average number of supplemental data links was 1.5. The maximum number of datasets associated with one article was 28; most of these were individual accessions for structures in the Protein Data Bank.

80.7% of the data links in the dataset corresponded to articles published in 2020, 2021, and 2022 (see Fig 1). The prevalence of newer links in the dataset is likely due to both the increased data sharing under modern funder and journal mandates as well as more publishers requesting that authors use data availability statements in articles [27]. A change in repository curation workflow in fall of 2022 meant that supplemental data links were not routinely captured after this point in time.

### How complete is this collection of supplemental data links?

Sampling of the published articles as compared to the metadata in CaltechAUTHORS found 21 of 450 sampled articles (4.7%) had supplemental data links and 120 of 450 articles (26.7%) were published with supplementary information; 17 of 450 articles (3.8%) had both. 11 of 450 articles (2.4%) had supplemental data links that were recorded correctly in CaltechAUTHORS. There were two types of error in the CaltechAUTHORS metadata: missing supplemental data links and accidentally captured links to supplementary information. 10 of 450 articles (2.2%) had links to shared data within the articles' data availability statements that were not recorded

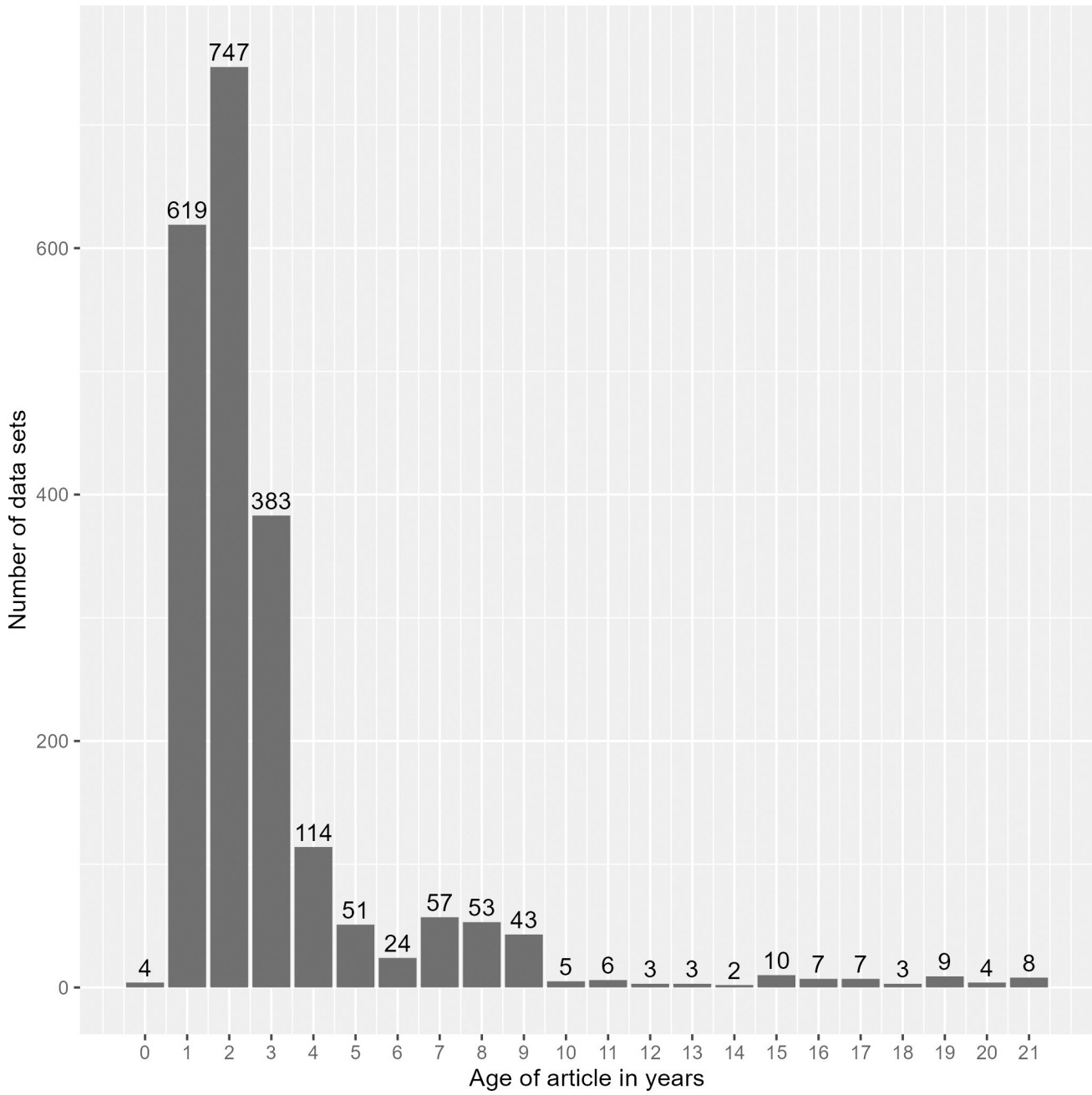

**Fig 1. Age of supplemental data links by publication year of the corresponding article.** This figure omits the 4 data supplemental data links prior to 2000.

in CaltechAUTHORS; and 6 of 450 articles (1.3%) had links to supplementary information that were incorrectly recorded in CaltechAUTHORS as supplemental data links. Based on sampling, the total supplemental data link curation error rate was 3.6%.

There was no noticeable pattern in the number of curation errors across the years sampled. Curation errors also appeared to be random with respect to publisher, as articles from several publishers were sometimes curated correctly while other articles from the same publisher had curation errors. The notable exception to this was for Cell Press, which started incorporating a

**Table 3. Top websites for sharing by URL.** This table includes websites with at least 10 shared datasets on the site.

| Website | Count of links |
| --- | --- |
| github.com | 152 |
| www.ncbi.nlm.nih.gov | 90 |
| data.caltech.edu | 33 |
| pds-geosciences.wustl.edu | 28 |
| osf.io | 26 |
| zenodo.org | 24 |
| www.ebi.ac.uk | 21 |
| spacephysics.princeton.edu | 20 |
| spdf.gsfc.nasa.gov | 19 |
| www.ccdc.cam.ac.uk | 18 |
| figshare.com | 18 |
| atmos.nmsu.edu | 15 |
| ars.els-cdn.com | 15 |
| www.sciencedirect.com | 14 |
| www-air.larc.nasa.gov | 14 |
| www.bco-dmo.org | 13 |
| fire.northwestern.edu | 13 |
| www.addgene.org | 11 |

"STAR Methods" section in its articles within the last decade; STAR Methods include data availability information among other content for reproducibility. Where supplemental data links from Cell Press appeared in data availability statements, they were curated as normal, but when data links appear within STAR Methods sections, they were not added to CaltechAUTHORS, resulting in incomplete inclusion of supplemental data links from this publisher. Based on sampling, the collection of supplemental data links analyzed in this article may be missing about half of the actual shared data links. Given the substantial size of the dataset and the fairly random selection of supplemental data links, the analysis in this article still provides a useful estimate of the continued availability of links to shared data despite being incomplete.

Sampling also demonstrated the accidental inclusion of some supplementary information in the dataset of supplemental data links. The sample dataset found that 6 of the 17 recorded supplemental data links in CaltechAUTHORS were inadvertently supplementary information. However, a corresponding estimate of one-third of the analyzed dataset being links to supplementary information is an overestimate when compared with other observations about the analyzed collection of links. Looking at the most common domains for shared data in Tables 3 and 4 shows only one publisher website, Science Direct, on a list otherwise full of data repositories. There are certainly some links to supplementary information hosted on less frequently used websites that are not shown in Tables 3 and 4, but they are not the vast majority of the links analyzed here. As it was not possible to filter out supplementary information links from the dataset, any results that center on links to supplementary information are called out and discussed in this analysis.

## Where are institutional authors sharing their research data? (RQ1)

Caltech researchers shared data on 513 different websites. This number generally does not include sites that mint DOIs, which are reported below; note that there is overlap between the lists of URL websites and DOI-minting organizations, as researchers sometimes reported a

**Table 4. Top data repositories for sharing, organized by administrator of the DOI prefix.** This table includes repositories with at least 10 shared datasets on the site.

| DOI prefix owner | Count of links |
|---|---|
| CaltechDATA | 156 |
| Zenodo | 139 |
| Figshare | 58 |
| Worldwide Protein Data Bank | 56 |
| Iris | 40 |
| Unavco | 29 |
| Mendeley | 25 |
| Dryad | 23 |
| OSF | 21 |
| Public Library of Science (PLoS) | 21 |
| Global Dataverse Community Consortium | 20 |
| EOSDIS | 14 |
| Caltech Library | 13 |
| ORNL Environmental Sciences Division | 10 |

dataset's URL instead of the preferable DOI when publishing an article. The websites with 10 or more supplemental data links are reported in Table 3. GitHub, a software sharing repository, is the most common website for sharing, likely because researchers are sharing data alongside software there. Several popular data repositories also appear on this list, such as OSF and Zenodo, even though they mint DOIs.

Caltech researchers shared data with 92 different organizations that mint DOIs. DOI prefix owners are given in Table 4 for organizations with at least 10 supplemental data links. Note that some organizations mint DOIs under multiple prefixes using the same owner name (these are grouped together in this analysis), and other organizations mint DOIs under different prefixes with different owner names (these are not grouped together here). For example, Caltech Library maintains multiple prefixes for the data repository CaltechDATA and several campus research projects, resulting in separate entries in Table 4 despite data being in the same repository.

172 (23.1%) of the 744 DOIs corresponded to prefixes minted by Caltech and its affiliated research groups. Between URLs and DOIs, links to the CaltechDATA data repository represented 9.5% (205) of all supplemental data links.

## What data is no longer available? (RQ2.1)

All of the supplemental data links that were URLs or DOIs (2086 links total)–but not FTP links, badly formatted URLs, or links to a non-HTML file type (80 links total)–were scraped algorithmically using R. 152 (7.3%) of the 2,086 links failed to scrape, of these 136 (6.5%) were URLs and 16 (0.8%) were DOIs. After testing links by hand in Chrome, 92 (4.4%) of the 2086 links still did not resolve– 79 (3.8%) URLs and 13 (0.6%) DOIs (see Table 5). 4 URLs did resolve to a webpage but asked for a login in order to see the data; these links were counted as resolving even though the data was not openly available.

The 21 FTP links and 56 links to non-HTML file types were also checked by hand. The 3 URLs that were identified in the analysis as badly formatted were corrected by hand at this point before testing. 26 (32.5%) of this group of 80 links did not resolve, with details given in Table 5.

In total, 118 (5.4%) of the 2166 supplemental links were no longer available. Of the link types with at least 10 links to test, DOIs were least likely to be unavailable (1.7% loss) and PDF

**Table 5. Summary of links that are no longer available, broken down by link type.**

| Link Type | Links that Fail to Resolve | Total Links of that Type | Percent of Fail Links of that Type |
|---|---|---|---|
| URL | 79 | 1342 | 5.9% |
| DOI | 13 | 744 | 1.7% |
| FTP | 5 | 21 | 23.8% |
| ZIP | 7 | 16 | 43.8% |
| PDF | 7 | 12 | 58.3% |
| DOCX | 2 | 9 | 22.2% |
| GZ | 3 | 7 | 42.9% |
| XLSX | 0 | 5 | 0.0% |
| Badly formatted URL | 1 | 3 | 33.3% |
| TXT | 0 | 3 | 0.0% |
| DOC | 0 | 2 | 0.0% |
| IMG | 0 | 1 | 0.0% |
| ZIPR | 1 | 1 | 100.0% |

files were most likely to no longer be available (58.3% loss). The PDF loss rate is complicated by the fact that many of these links point to supplementary information instead of shared datasets; this means the link information was recorded erroneously in CaltechAUTHORS (supplementary information files should have been uploaded to the repository instead of recorded as supplemental data links). As an estimated quarter of Caltech-authored articles contain supplementary information, a significant amount of which is formatted as a PDF, the continued availability of PDF data on the internet merits further study for better quantification.

### How does data availability trend over time? (RQ2.2)

Examining links that were unavailable by article publication year (see Fig 2), a pattern emerges where older links were more likely to be unavailable than newer links. The number of links tested varied between publication years–with fewer supplemental data links tested for older publications–so only data from 2014–2022 is plotted and fit in Fig 2, as these years had at least 10 links to test. Modeling the relationship between availability of the datasets and age of the article using a Poisson regression, the odds of data being available was found to reduce by 2.6% (odds ratio = 0.974 [.950-.998, 95% CI], $p < 0.05$) for each year after the article is published.

### Are there specific sites where data is more likely to disappear? (RQ2.3)

Tables 6 and 7 list websites and repositories, respectively, where at least two datasets have gone missing. ScienceDirect was the website with the most missing data; half of ScienceDirect's 14 links were missing. Many other journal websites appear in Table 6, suggesting that journal-hosted supplementary information is not a stable method of data sharing. As supplementary information is so prevalent (see previous discussion about PDF Links) and was only incidentally tested in this analysis, the continued availability of journal-hosted supplementary information merits further investigation. GitHub also appears at the top of Table 6, with 5 missing datasets, suggesting that it is also not an ideal place to store data in a sustainable way.

### Discussion

This research has several limitations. First, this analysis is not a full measure of "data rot" as it did not check for content drift (i.e. that the harvested webpage actually represents the data

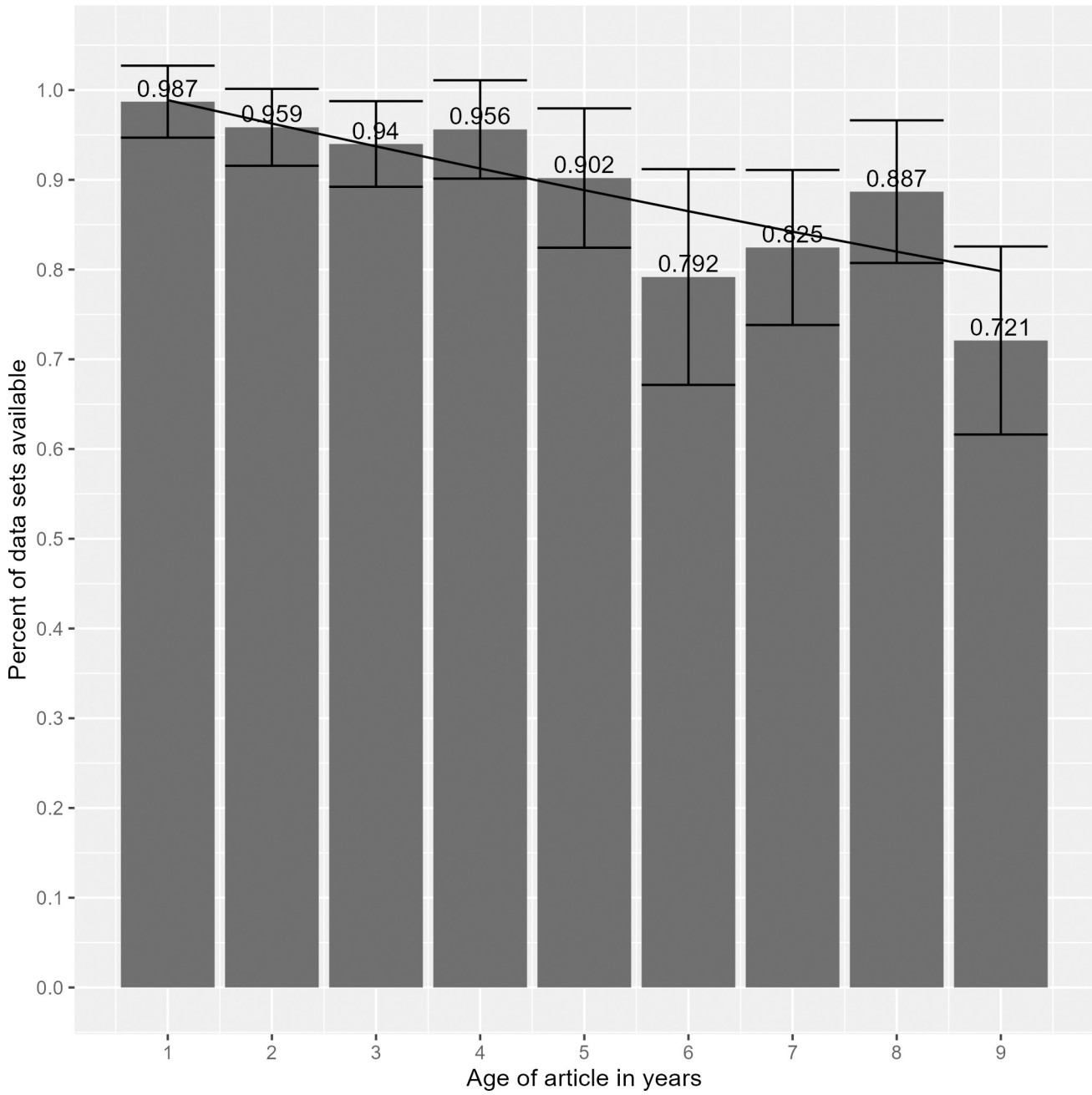

**Fig 2. Percentage of supplemental data links that are no longer available, charted by article publication year.** The figure only includes data from 2014 to 2022, as those years all had at least 10 links to test.

shared by the article). This would entail checking thousands of webpages against article meta-data and is beyond the scope of this analysis, but would be a useful future analysis. Another limitation is that this methodology is not reproducible at other universities unless they have significantly invested in tracking institutional research outputs. It is also important to note that this data source is not perfect for two reasons. First, the inclusion criteria for this dataset leverages messy text in the repository's "related link description" metadata field, meaning some included links may not correspond to actual datasets and other data links may have been

**Table 6. Websites with at least 2 supplemental data links that are no longer available.**

| Website | Count of links |
|---|---|
| www.sciencedirect.com | 7 |
| github.com | 5 |
| ppi.pds.nasa.gov | 4 |
| www.plantphysiol.org | 4 |
| co2.jpl.nasa.gov | 3 |
| www.geosociety.org | 3 |
| www.plantcell.org | 3 |
| mccarthy.well.ox.ac.uk | 2 |
| www.lncRNA.caltech.edu | 2 |
| www.jimmunol.org | 2 |
| www.pdb.org | 2 |
| www.nature.com | 2 |
| diabetes.diabetesjournals.org | 2 |
| iopscience.iop.org | 2 |
| avdc.gsfc.nasa.gov | 2 |
| archive.stsci.edu | 2 |
| www2.physik.uni-kiel.de | 2 |

inadvertently excluded from analysis. Second, due to repository curation errors, this analysis is missing a significant number of supplemental data links and accidentally includes links to supplementary information. That said, the accidental inclusion of links to supplementary information provides evidence that data shared in a data repository using a DOI is more stable than data hosted by a journal.

This analysis found data availability measurements comparable to other studies in the literature. The most similar study by Federer, et al. found a 78% and 98% hand-retrieval measurement for URLs and DOIs for shared data, respectively [20]. The 98% success measurement for DOIs is equivalent to the 1.7% unavailability measurement determined here. Federer's 78% value for URLs was calculated from 5-to-7-year-old papers. In this analysis, an equivalent value is the range of 9.8%-20.8% unavailability for papers between 2016–2018 (this range almost exclusively represents failed URL and file retrievals rather than failed DOI or FTP links). This analysis is near Federer's measured value, despite testing fewer older links. This analysis's results can also be compared to findings from Pepe, et al. which looked at availability of data shared with astronomy articles. Pepe found a similar decrease in data availability over time, though with larger values than measured here, likely due to differences in inclusion criteria for tested links; Pepe's data was not modelled to determine the rate of data loss over time. Federer did find a small difference–though not a statistically significant one–between data availability over their 3-year window of analysis, though Vines, et al. found that data availability dropped off over time by 17% per year [23]. However, the Vines study contacted authors for their data rather than harvesting web-accessible data algorithmically. The actually availability of data over time is likely somewhere between the two, meaning the data availability drop off of 2.6% per year measured in this article is a reasonable estimate.

**Table 7. Repositories with at least 2 DOIs that fail to resolve.**

| DOI prefix owner | Count of links |
|---|---|
| Mendeley | 3 |
| ISTIC | 3 |

The results presented here bring to light several problems with shared URLs. First, researchers have clearly shared URLs from repository like OSF, Zenodo, and Figshare (see Table 3) which have the ability to mint DOIs. This means that researchers are either not aware of shareable DOIs or choosing to share the dataset's URL instead an available DOI. This is consistent with previous work by Van de Sompel, et al., who found that researchers regularly cite URLs instead of available DOIs when referencing scholarly articles [28]. Given the evidence here and elsewhere that DOIs are more stable and the recommendation from federal funding agencies–such as NIH [5] and even higher-level guidance from OSTP [6]–that persistent identifiers like DOIs are preferred for shared data, this is a gap that needs to be addressed through researcher education and guidance. Second, many researchers are sharing links directly to files rather than links to a specific webpage, which elides important metadata that may accompany the file. These links also appear to be the least stable, as compared to URLs and DOIs–another problem that merits addressing. Finally, over 10% of shared URLs point to website homepages, which is not helpful information for tracking down the exact dataset used in an analysis. Altogether, there is clear need for librarians, data curators, journals, and others to educate researchers on best practices for formatting links for data sharing. This guidance should be that: DOIs are preferred, use a DOI that points to a repository record rather than a link to a specific file, and the DOI should point to the specific dataset used in the analysis instead of an entire database.

Another form of link that was found to be particularly problematic in this analysis was data shared as PDFs. There are many reasons not to use PDFs for sharing data, the chief of which is that data shared in PDFs is dead data that can be almost impossible to extract and reuse. PDF data is frequently found as supplementary information on journal websites, but journals often migrate platforms which can break URLs; this is evidenced by the almost 60% of PDF data that was not found in this analysis. Thankfully, the latest data sharing recommendations, such as those stipulated by NIH [5], mandate that researchers share usable data in a data repository rather than relegating data into a PDF on a publisher website. Given the high prevalence of supplementary information accompanying published articles, there is significant work to be done to shift journals away from supplementary PDFs and have researchers share data in a data repository instead.

A final positive outcome of this analysis is the finding that there is high usage of the institutional data repository, CaltechDATA. The data repository is well supported by the library, with conscious effort to make upload and DOI minting easy for researchers. This has clearly translated into solid uptake by Caltech researchers. This evidence proves the value of institutional data repositories to meet researcher need where other disciplinary repositories may not be available.

## Conclusion

Despite increases in data sharing, data is sometimes unavailable for download and reuse due to data rot. Some of this is not the fault of the researcher (such as when DOIs are not properly maintained) but there are improvements that researchers can make to ensure their data remains available. In particular, researchers should follow best practice guidance to deposit data into a data repository and share it using a permanent identifier. This will ensure that shared data is as sustainable as possible and available for future researchers.

## Acknowledgments

The author thanks Tom Morrell for downloading metadata from the CaltechAUTHORS repository, recommending CyberDuck for testing FTP links, and providing feedback on a

copy of the manuscript draft. Thanks to George Porter for answering questions about the specifications of the CaltechAUTHORS metadata. Thank you to George, Tony Diaz, and others in the Caltech Library who have dedicated a huge amount of time and effort to populating the CaltechAUTHORS repository and ensuring the quality of its metadata.

## Author Contributions

**Conceptualization:** Kristin A. Briney.

**Data curation:** Kristin A. Briney.

**Formal analysis:** Kristin A. Briney.

**Investigation:** Kristin A. Briney.

**Methodology:** Kristin A. Briney.

**Software:** Kristin A. Briney.

**Validation:** Kristin A. Briney.

**Visualization:** Kristin A. Briney.

**Writing – original draft:** Kristin A. Briney.

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
