## [Decision Letter · Decision Letter 0]

26 Sep 2023

PONE-D-23-20069Measuring Data Rot: An Analysis of the Continued Availability of Shared Data from a Single UniversityPLOS ONE

Dear Dr. Briney,

Thank you for submitting your manuscript to PLOS ONE. After careful consideration, we feel that it has merit but does not fully meet PLOS ONE’s publication criteria as it currently stands. Therefore, we invite you to submit a revised version of the manuscript that addresses the points raised during the review process.

Given the comments by the reviewers, a major revision seems to be in order. Especially the second reviewer raises a number of issues that should be addressed. In particular, the reviewers raise several questions around the representativeness of the sample from Caltech that should be addressed.

In addition to the issues identified by the reviewers, I would like to suggest one additional methodological change. At the moment, the estimate of the 2.4% rate of disappearance per year is estimated using a linear regression. However, if each year 2.4% of the data disappears, then you would expect 1 - (1 - 0.024^t) data to have disappeared after t years, not t * 0.0024. That is, stated succinctly, it would be better to use survival analysis for the estimation of this yearly disappearance instead of a linear regression.

We look forward to receiving your revised manuscript.

Kind regards,

Vincent Antonio Traag, Ph.D.

Academic Editor

PLOS ONE

Journal Requirements:

Reviewers' comments:

Reviewer's Responses to Questions

**Comments to the Author**

1. Is the manuscript technically sound, and do the data support the conclusions?

Reviewer #1: Yes

Reviewer #2: Yes

2. Has the statistical analysis been performed appropriately and rigorously? 

Reviewer #1: Yes

Reviewer #2: N/A

3. Have the authors made all data underlying the findings in their manuscript fully available?

Reviewer #1: Yes

Reviewer #2: Yes

4. Is the manuscript presented in an intelligible fashion and written in standard English?

Reviewer #1: Yes

Reviewer #2: Yes

5. Review Comments to the Author

Reviewer #1: Thank you for your work, data sharing is a very timely and important topic and it is encouraging to see more research on it.

I believe this study is clear and well structured, it properly documents all the steps in the analysis and allows for full replication of the results. The discussion of the related art is comprehensive and direct comparisons are made where appropriate. The provided summary statistics and figures are appropriate for the envisaged level of analysis. Furthermore, the study concludes by highlighting best practices which should be more broadly adopted.

With respect to limitations, I see two main ones which I would encourage the authors to consider (as they partially do in the discussion):

- The data sample is from Caltech, therefore a single university with a certain disciplinary focus. While this might not be a problem per se, it would be interesting to know something more about what articles are included in the sample. For example, the disciplinary breakdown and the most represented venues. This would help the reader getting a better sense of where the results apply most (as well as ideas for future work in terms of extending the study considering complementary samples).

- I am not sure to what extent the data on related links to datasets can be considered to be fully correct and, if not, how this might effect the results. This is also something that the authors mention, yet I am left wondering about this: would it be possible to strengthen the study by addressing this question more systematically? For example, a small random sample of entries could be manually checked to understand what might be the share of missing data in the related links to datasets field.

I hope my comments can help improve and finalise this useful contribution.

Reviewer #2: The author presents a study exploring the data sharing behaviours of researchers at a single university by looking at the “supplemental data links'' included within the metadata of publications within the university’s institutional repository. Overall, the article is well written, uses methods appropriate to the research questions, and presents findings which will be of interest to the research data community. There are a few open questions and points for revision, which I outline below.

INTRODUCTION/BACKGROUND

Currently there is not a separate literature review or background section. I recommend splitting the current introduction into two sections and perhaps adding a bit more content in terms of the literature reviewed.

There are also a few areas where the referencing could be improved, or where it was unclear which reference was being used to substantiate a statement.

On page 3, line 36 - Are there references to include for the concepts of link rot and content drift?

On page 4, line 56 - It is unclear which (if any) of the preceding references are used to support the statement that “Gaining credit for sharing and support for sharing would make researchers more likely to share data but there is still a perceived risk in sharing.”

The authors provide in text citations to the R packages used in the analysis, which is great. Should these citations also appear in the reference list?

The author may also want to discuss (either in the literature review or in the discussion) how their work relates to an earlier study about data rot in the field of astronomy [1]. The combination of methods used in the astronomy study may also be interesting to the author in terms of developing future research directions to explore the motivations behind the behaviours which were observed.

METHODS

The methodology is clearly described. I wonder, though, if the author could include more detail about how the repository team “captures” the articles produced by the university. Does the repository staff perform literature searches to populate the repository? Do researchers voluntarily deposit articles? Essentially, I am wondering how complete the corpus of articles is, compared to the output of the university to understand how representative the data sharing behaviours are of researchers at the institution.

RESULTS

I find it particularly interesting that data sharing via journal supplemental material and GitHub were the least sustainable mechanisms, and that researchers link to files which essentially bypasses metadata which could be helpful in making sense of the data.

Reading the results, I had a few questions. A minor point is that on Line 153, the author mentions an article with 28 associated datasets - do they have any more insight into why this article has so many associations? I also wonder about the change in workflow which they mention in 2022; has this been changed again, so that the repository can capture information in the future?

I also wonder about the disciplinary distribution of articles/data in the corpus which was analyzed, given the tendency to use GitHub. It would be interesting and provide context to the results if the author could provide more information about disciplines here.

I also recommend that the author consider changing the ordering of the x-axes in both of the figures so that years increase rather than decrease. It is counter-intuitive and more difficult to interpret the data in the current presentation.

DISCUSSION

The author raises an interesting finding in Line 279 - that researchers are either choosing not to mint DOIs or are sharing data via a URL. The last option seems very likely to me, given existing work documenting a similar behavior for how researchers share academic articles with each other, as explored by Herbert Van de Sompel and Martin Klein. (The work from these authors could also be used in terms of content drift and link rot more broadly). It would be interesting for the author to make this comparison.

The author also draws on their findings to call for a need to educate researchers. I wonder what other recommendations can be made for other stakeholders, e.g. for curatorial staff.

As a minor point, I agree with the statement that there appears to be a solid uptake in use of the repository by Caltech researchers (Line 304). I wonder though, what evidence the author is using to back up the statement that there are no disciplinary repositories for the researchers who use CaltechDATA. It is an interesting question to think about who uses institutional repositories (as in which disciplines) and for which data. Perhaps the author has more to add to this statement.

MINOR ERRORS/TYPOS/CLARITY ISSUES

On P.3 , line 53 - “is” should be “if”

On p.11, Line 201, p.11 there appears to be a reporting error in the statement that “...of these 136 (6.5%) were URLS and 16 (0.8%) were URLs” Are both percentages for URLs?

On line 249, instead of stating that “this article is not reproducible”, it may be better to say that this methodology is not reproducible?

I suggest deleting “So” at the beginning of the sentence on Line 268.

On p.4, line 59: The statement “Many researchers, however, are actively sharing their data under data sharing requirements from journals” seems to contradict the following paragraph regarding problems with data availability statements.

Line 74 is also unclear; I wonder if the following statistic was perhaps copied over incorrectly from the Federer paper. “Federer found that 80% of data resources could be retrieved automatically, a percentage that went up to 78% for URLs and 98% of DOIs resources when testing links by hand.”

I would also encourage the author to clearly define the terms “supplemental datasets” and “supplemental data links” as early as possible, as these terms may not be intuitive to every reader.

REFERENCES

Pepe, A., Goodman, A., Muench, A., Crosas, M., & Erdmann, C. (2014). How do astronomers share data? Reliability and persistence of datasets linked in AAS publications and a qualitative study of data practices among US astronomers. PLoS ONE, 9(8). https://doi.org/10.1371/journal.pone.0104798

6. PLOS authors have the option to publish the peer review history of their article (what does this mean?). If published, this will include your full peer review and any attached files.

Reviewer #1: No

Reviewer #2: No

---

## [Author Response · Author response to Decision Letter 0]

6 Mar 2024

I have attempted to address all of the reviewer comments and have listed the updates here. Due to the significant number of changes, they are roughly listed according to the article section in which they appear for ease of tracking:

Introduction

- I separated the introduction and the literature review.

- I added definitions of “supplemental data link” and “supplementary information”. I have tried to consistently use these two terms throughout the manuscript.

Literature review

- I added references to work done by Herbert Van de Sompel and Martin Klein on reference rot and content drift, alongside a recently published article about the preservation of published articles by Martin Eve.

- I added reference to Pepe, et al. article about availability of data from astronomy articles. I also cited data from this article in the discussion for comparison to findings in this analysis.

- I clarified which Borgman and Tenopir citations support the claim for making researchers more likely to share data.

Methodology

- I added information on the process by which we search for Caltech-authored articles to add to the institutional repository.

- I added information as to why we stopped curation supplemental data links in 2022 (to prepare for a repository upgrade in 2023).

- I added information on the disciplinary breakdown of Caltech-authored articles, pulling data from Web of Science.

- I sampled article metadata from the institutional repository and compared this information in the published articles to gather an estimate of the accuracy and completeness of the collection of supplemental data links used in this analysis. Sampling information was reported in the results section. Findings impacted my error calculations and the discussions about supplementary information and PDF links throughout the article.

- I added citations to the R packages as references. I removed the mention of the R package “utils” as I had removed use of that from the final code.

Results

- I added a note about the one article in the dataset that contains 28 links to supplemental data (most of the links go to individual structure accessions in Protein Data Bank).

- I changed the x-axes of both figures to show the age of the article rather than the year published, so that newer articles are now on the left and older articles are on the right.

- I flipped the y-axis on figure 2 by plotting the availability of data rather than the unavailability for consistency with corresponding studies.

- I updated the fit model for figure 2 to be a Poisson regression, as was done in a corresponding study by Vines, et al. This fit includes a confidence interval and p-value.

- I redid all of the error calculations, fit modelling, and figures in R so that everything is reproducible and transparent.

Discussion

- I updated the discussion about limitations of the dataset to better reflect the results of sampling the quality of the metadata in the repository.

- I added a reference to work by Van de Sompel, et al. showing that researchers use URLs instead of DOIs when citing scholarly articles.

- I added a recommendation that journals should shift away from supplementary information in PDFs in favor of having researcher put data in to a data repository.

- I did not do an analysis of the disciplines of data in the institutional data repository, CaltechDATA, to see where it might be filling gaps in the landscape of disciplinary repositories. This is beyond the scope of this article.

Data availability

- I updated the links to the shared data and code as the repository generates new DOIs for each version of the content.

---

## [Decision Letter · Decision Letter 1]

14 May 2024

PONE-D-23-20069R1Measuring Data Rot: An Analysis of the Continued Availability of Shared Data from a Single UniversityPLOS ONE

Dear Dr. Briney,

Thank you for submitting your manuscript to PLOS ONE. After careful consideration, we feel that it has merit but does not fully meet PLOS ONE’s publication criteria as it currently stands. Therefore, we invite you to submit a revised version of the manuscript that addresses the points raised during the review process.

Your revisions satisfied both reviewers, and so, in principle, we could accept your publication. However, the second reviewer identified some minor errors, and I wanted to offer you the opportunity to revise those before accepting the final version, hence the decision to ask for a minor revision.

We look forward to receiving your revised manuscript.

Kind regards,

Vincent Antonio Traag, Ph.D.

Academic Editor

PLOS ONE

Journal Requirements:

Additional Editor Comments:

Reviewers' comments:

Reviewer's Responses to Questions

**Comments to the Author**

1. If the authors have adequately addressed your comments raised in a previous round of review and you feel that this manuscript is now acceptable for publication, you may indicate that here to bypass the “Comments to the Author” section, enter your conflict of interest statement in the “Confidential to Editor” section, and submit your "Accept" recommendation.

Reviewer #1: All comments have been addressed

Reviewer #2: All comments have been addressed

2. Is the manuscript technically sound, and do the data support the conclusions?

Reviewer #1: Yes

Reviewer #2: Yes

3. Has the statistical analysis been performed appropriately and rigorously? 

Reviewer #1: Yes

Reviewer #2: I Don't Know

4. Have the authors made all data underlying the findings in their manuscript fully available?

Reviewer #1: Yes

Reviewer #2: Yes

5. Is the manuscript presented in an intelligible fashion and written in standard English?

Reviewer #1: Yes

Reviewer #2: Yes

6. Review Comments to the Author

Reviewer #1: Thank you for addressing my remarks, I have no further comments.

Reviewer #2: Thank you for your work in revising this manuscript. The added detail, changes to the methodology, analysis, and situation in the literature has improved the manuscript. I noted only a few minor errors (typographical errors mostly), one place where an added reference would strengthen the article, and a question about the reporting of the data curation error rate.

P.3, Line 43: text should be “It can BE difficult…”

P.3, Line 50: the text in parantheses should be a new sentence

P.11, Line 215: same problem as above

P.12, Line 222: a reference is needed for this claim: “more publishers requesting that authors use data availability statements in articles instead of noting shared data as a citation or footnote.” Can the authors cite a study that shows that publishers request the use of data availability statements more often than requesting data citations?

P. 13, Line 242: Should it be clarified that the data curation error rate is calculated only based on the sample?

7. PLOS authors have the option to publish the peer review history of their article (what does this mean?). If published, this will include your full peer review and any attached files.

Reviewer #1: No

Reviewer #2: No

---

## [Author Response · Author response to Decision Letter 1]

15 May 2024

I have addressed all of the comments and have listed the updates here. The changes are as follows, with the request and how I modified the manuscript in the subsequent bullet point:

Request: P.3, Line 43: text should be “It can BE difficult…”

• I updated the text as requested.

Request: P.3, Line 50: the text in parantheses should be a new sentence

• I made the parenthetical into a new sentence.

Request: P.11, Line 215: same problem as above

• I updated the parenthetical to be part of the same sentence following a semicolon.

Request: P.12, Line 222: a reference is needed for this claim: “more publishers requesting that authors use data availability statements in articles instead of noting shared data as a citation or footnote.” Can the authors cite a study that shows that publishers request the use of data availability statements more often than requesting data citations?

• I removed the following portion of the text – “instead of noting shared data as a citation or footnote” – and added citations to show 

the increase in published data availability statements. This update better reflects the point I’m making about increased data availability statements (as opposed to the explicit method of how data is shared in articles).

Request: P. 13, Line 242: Should it be clarified that the data curation error rate is calculated only based on the sample?

• I edited the sentence to reflect the calculated error rate is based on sampling.

---

## [Editor Report · Decision Letter 2]

20 May 2024

Measuring Data Rot: An Analysis of the Continued Availability of Shared Data from a Single University

PONE-D-23-20069R2

Dear Dr. Briney,

We’re pleased to inform you that your manuscript has been judged scientifically suitable for publication and will be formally accepted for publication once it meets all outstanding technical requirements.

Kind regards,

Vincent Antonio Traag, Ph.D.

Academic Editor

PLOS ONE

---

## [Editor Report · Acceptance letter]

21 May 2024

PONE-D-23-20069R2 

PLOS ONE

Dear Dr. Briney, 

I'm pleased to inform you that your manuscript has been deemed suitable for publication in PLOS ONE. Congratulations! Your manuscript is now being handed over to our production team.

Kind regards, 

on behalf of

Dr. Vincent Antonio Traag 

Academic Editor

PLOS ONE